# Genetic Diversity and Connectivity of *Ocypode ceratophthalmus* in the East and South China Seas and Its Implications for Conservation

**DOI:** 10.3390/biology12030437

**Published:** 2023-03-12

**Authors:** Feng Zhao, Yue Liu, Zihan Wang, Jiaying Lu, Ling Cao, Cong Zeng

**Affiliations:** School of Oceanography, Shanghai Jiao Tong University, Shanghai 200030, China

**Keywords:** population structure, East China Sea, South China Sea, horn-eyed ghost crab, conservation

## Abstract

**Simple Summary:**

This study investigated the genetic diversity and connectivity of 15 *Ocypode ceratophthalmus* populations in the East and South China Seas based on two genetic markers. The results showed that *O. ceratophthalmus* had a high genetic diversity among all collected populations, and an insignificant population structure was observed by a hierarchical analysis of molecular variance and fixation index. Additionally, Migrate-n revealed high historical gene flow and migration rates among populations. The results of this study could inform the construction and management of marine protected areas in the East and South China Seas.

**Abstract:**

The East and South China Seas are rich in marine resources, but they are also under great pressure from climate change and human activities. Maintaining diversity and connectivity between communities is thought to be effective in mitigating these pressures. To assess the diversity and connectivity among the populations of *Ocypode ceratophthalmus* in the East and South China Seas, 15 populations from or near 15 marine protected areas in the two seas were studied using *COI* and *D-Loop* as genetic markers. The results showed that *O. ceratophthalmus* populations had high diversity, and the results of a hierarchical analysis of molecular variance and fixation index found that there were no significant genetic structures among these populations. High historical gene flow and high migration rates were further observed among populations by Migrate-n. Furthermore, the *COI* sequences further showed the asymmetric migration rate with a higher migration rate from south to north than from north to south. This information could provide recommendations for the management of marine protected areas in the East and South China Seas.

## 1. Introduction

The East and South China Seas are important areas for the utilization of marine resources and marine conservation. Currently, 12,933 tropical and subtropical species in the East China Sea (ECS) have been recorded, of which half (48%) were endemic [1], and the ECS was also an important fishery for the world, contributing about 6 million tons of catches annually [2]. The South China Sea (SCS), adjacent to the ECS, is regarded as the hotspot of tropical biodiversity in shallow seas, with 450 species of coral (about 7% of the world’s total coral reef areas), 2 million hectares of mangrove (12% of the world’s total mangrove areas), 1027 species of fish, 91 species of shrimp and 73 species of cephalopods [3]. Furthermore, the SCS provides 6 billion tons of catch annually, accounting for 10% of the world’s total catch [3]. Nevertheless, with the rapid economic development in the East and South China Seas, the regions were also facing threats, such as overfishing, biodiversity decline and habitat degradation, which urgently need to be strengthened for conservation [3]. The marine protected area (MPA) was considered the core initiative of marine biodiversity conservation [4], and MPAs were established to protect the ECS and SCS 20 years ago by the Chinese government. So far, 22 national-level MPAs and more than 50 local-level MPAs have been established in the ECS, and more than 20 national-level MPAs have been set up in the SCS [5]. Most of these protected areas were usually designated by local governments, and these bottom-up designations may lack systematic planning [6].

Protecting biodiversity as a whole required not only the design of protected areas with reasonable functions, but also the consideration of species migration among different habitats on a larger scale, such as the establishment of several protected areas with the same functions to form an MPA network. Previous studies have also revealed that the MPA network could effectively prevent or mitigate the negative impacts of biodiversity [7], which required connectivity to promote population persistence in marine ecosystems [8]. The Convention on Biological Diversity has recognized connectivity as a fundamental principle in the planning of the MPA network. Additionally, it was requisite to determine the boundaries of ecosystems based on connectivity in the ecosystem-based marine reserve management system [9]. Therefore, it is important to understand the connectivity in these two seas and that connectivity barriers may not be present in these two seas.

The ECS belongs to the East China Sea Ecoregion in the Warm Temperate Northern Pacific Marine Biogeographic province, and the SCS belongs to the Gulf of Tonkin Ecoregion, Southern China Ecoregion and South China Sea Oceanic Islands Ecoregion in the South China Sea Marine Biogeographic province [10]. Whether the transition in different ecoregions will have an impact on population connectivity remains to be researched, and this question relates to whether the protected areas between the two seas are constructed as a network of protected areas or separately. At present, studies on this issue do not have the same results. On the one hand, high connectivity reports were increasing in coastal species, which span different ecoregions. For example, *Atrina pectinate* [11], *Nibea albiflora* [12], *Octopus ovulum* [13], *Parasesarma affinis* [14], *Periophthalmus modestus* [15], *Siphonaria japonica* [16] and *Thamnaconus hypargyreus* [17] had high connectivity among populations in both ECS and SCS. On the other hand, it has been reported that there was limited connectivity between populations in the ECS and populations in the SCS, such as *Chelon haematocheilus* [18], *Eleutheronema tetradactylum* [19], *Epinephelus akaara* [20], *Pagrus major* [21] and *Scomber japonicus* [22]. These inconsistent findings, which could be caused by limited spatial coverage and insufficient sampling (usually less than 10 populations, see summary in Appendix A), require further research on the connectivity of the two seas.

Horn-eyed ghost crab (*Ocypode ceratophthalmus*), a species widely distributed in the East and South China Seas, has a planktonic larval stage and a settled adult stage like most marine invertebrates. Furthermore, *O. ceratophthalmus* is expected to suffer less human disturbance due to its low economic value, and it is therefore considered to be better to reflect the diversity and connectivity patterns among protected areas [23]. Although the previous study did not find any genetic differences in the SCS population, it was tested in only three populations and with a limited sample [24]. To reveal its population structures and provide suggestions for the marine protection in the East and South China Seas, two molecular markers, *COI* and *D-Loop*, were employed in this study to assess the genetic connectivity among horn-eyed ghost crab populations in or near marine protected areas in the two seas. The results from this study were expected to inform the construction and management of marine protected areas in China.

## 2. Materials and Methods

### 2.1. Sample Collection and DNA Extraction

From September 2021 to April 2022, 8 populations of the horn-eyed ghost crab were collected from sandy beach in 8 marine protected areas. In addition, because not every MPA has the ghost crab habitat, 7 populations were collected in the adjoining sandy beach of 7 marine protected areas (Figure 1; Table 1). The morphological details of each group are presented in Appendix A.

The obtained samples were morphologically identified according to the previous study [25], samples were transported back to the laboratory, measurements of carapace width, carapace length, abdomen width, abdomen length, weight and gender were conducted [26]. Then, samples were stored in 95% ethanol at −20 °C until DNA extraction. About 20 mg of muscle tissue from the ambulatory leg was obtained to extract the total DNA by using MolPure^®^ Cell/Tissue DNA Kit (Yeason, Shanghai, Chian), and the DNA was stored at −20 °C and used as a template in the PCR reactions. PCR was implemented to amplify *COI* and *D-Loop* using the following primers, LCO1490 (5′-GGTCAACAAATCATAAAGATATTGG-3′) and HCO2198 (5′-TAAACTTCAGGGTGACCAAAAAATCA-3′) for *COI*, 13,323F (5′-GCGAATGCTGGCACAAACAT-3′) and 14,378R (5′-AGGGAGTGGTGCAATTCCAT-3′) for *D-loop*. The PCR amplification reaction included 25 μL 2× Hieff^®^ PCR Master Mix, 2 μL of upstream and downstream primers (10 μM), about 55 ng of template DNA, and double-distilled water was added to the total volume of 50 μL. Then, the PCR thermal cycling program was set as follows: denaturation at 94 °C for 5 min, followed by 35 cycles of denaturation for 94 °C 30 s, annealing at 55 °C for 30 s, elongation at 72 °C for 1 min; and the final extension step for 10 min at 72 °C. All PCR products were detected by 1.5% agarose gel electrophoresis. Finally, the PCR products were purified and sequenced by Tsingke Biotech Company (Shanghai, China).

**Figure 1 biology-12-00437-f001:**
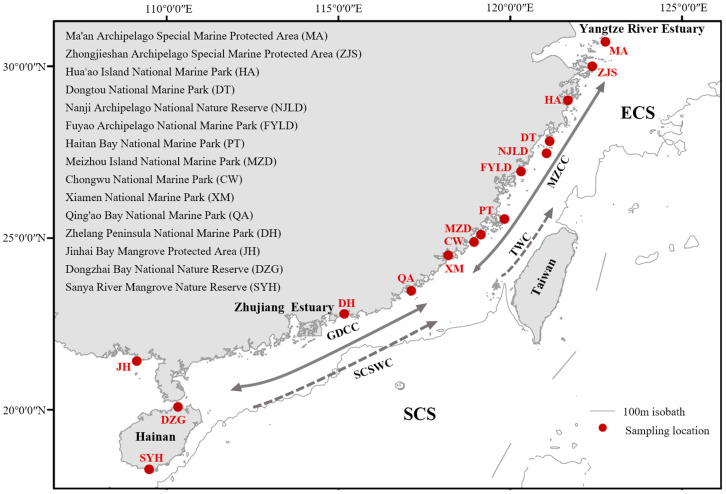
Sampling locations and ocean currents schematic in the studied area. TWC, Taiwan Warm Current; SCSWC, South China Sea Warm Current; MZCC, the coastal currents of Zhejiang and Fujian; GDCC, Guangdong coastal current. The direction of MZCC and GDCC varies seasonally, from south to north in summer and vice versa in winter [27], which is indicated by the solid line with bidirectional arrows in the figure.

### 2.2. Statistical Analyses of Genetic Data

All sequences were checked and edited using Geneious. Multiple sequences were aligned using the Geneious Alignment (Identity 1.0/0.0), and all alignments were checked visually. MEGA11 was used to evaluate the composition of the nucleotides. DnaSP v.4.0 was used to calculate the genetic diversity indices such as the number of haplotypes (h), haplotype diversity (Hd) and nucleotide diversity (π).

A map of haplotypes distribution was constructed using ArcGIS, R 4.2 and Arlequin 3.5, and minimum spanning networks were constructed using PopART 1.7. Hierarchical analysis of molecular variance (AMOVA) of the different levels was used to further examine the population structure. The AMOVA based on three levels included population level (MA, ZJS, HA, DT, NJLD, FYLD, PT, MZD, CW, XM, QA, DH, JH, DZG, SYH), region level (the East China Sea: MA, ZJS, HA, DT, NJLD, FYLD, PT, MZD, CW, XM; the South China Sea: QA, DH, JH, DZG, SYH) and ecoregion level (East China Sea Ecoregion: MZD, PT, FYLD, NJLD, DT, HA, ZJS, MA; Southern China Ecoregion: DH, QA, XM, CW; Gulf of Tonkin Ecoregion: JH, DZG, SYH). The significance of AMOVA was analyzed by 10,000 permutations. The fixation index (*F_ST_*) was employed to assess the pairwise genetic divergence between different populations and the significance was obtained by 10,000 permutations.

Migrate-n 3.6 was conducted to estimate the mutation-scaled migration rate M (M = m / μ, where m was the historical migration rate among populations and μ was mutation per generation) and mutation-scaled population size θ (θ = Ne × µ, where Ne was historical effective population size). The runs were recorded every 100 steps for a total of 1,000,000 long-chain Markov chain Monte Carlo (MCMC) steps and a burn-in of 1000. To improve the efficiency of the MCMC search, a static heating scheme with four different temperatures (1.0, 1.5, 3.0 and 1,000,000.0) was used. We inspected histograms of estimated θ and M posterior values to assess convergence. We calculated historical gene flow (Nm) by using the equation Nm = θ × M (when using mitochondrial gene).

## 3. Results

### 3.1. Population Genetic Diversity

A total of 253 *COI* sequences (538 bp, OP989704-956) and 95 *D-Loop* sequences (611 bp-612 bp, ON504951-995) were obtained from 15 populations of horn-eyed ghost crab. For *COI* sequences, the average contents of A, T, G and C were 26.6%, 34.6%, 16.8% and 22.0%, respectively. A total of 50 polymorphic sites were detected, including 24 singleton variable sites and 26 parsimony informative sites. Haplotype diversity ranged from 0.80 to 0.98, and nucleotide diversity ranged from 0.0022 to 0.0054 (Table 2). When the 15 populations were considered as a metapopulation, the haplotype diversity was 0.87, and the nucleotide diversity was 0.0036 (Table 2).

The average contents of A, T, G and C were 42.7%, 32.6%, 9.6% and 15.0%, respectively, according to the results based on *D-Loop* sequences. A total of 116 polymorphic sites were found, including 89 parsimony informative sites and 27 singleton variable sites. The genetic diversity parameters showed that nucleotide diversity ranged from 0.0192 to 0.0379, and haplotype diversity ranged from 0.90 to 1.00 (Table 2). When the 15 populations were considered as a whole, the nucleotide diversity was 0.0279, and the haplotype diversity was 1.00. In conclusion, both genetic markers demonstrated high genetic diversity in horn-eyed ghost crab populations.

### 3.2. Haplotype Analysis

The *COI* sequences identified 62 haplotypes (Hap 1-62), including 24 shared haplotypes and 38 exclusive haplotypes. The frequencies of 62 haplotypes varied widely; the combined frequencies of Hap 1 and Hap 2 were as high as 49.40%, with the highest frequency of Hap 2 (26.48%), followed by Hap 1 (22.92%), while the remaining haplotypes were less frequent. Two haplotypes (Hap 1-2) were shared by 15 populations (Figure 2), while common haplotypes (Hap 1-2, Hap 7, Hap 10 and Hap 13) also existed in populations that were far apart (up to 1000 km), according to the geographic distribution of haplotypes (Figure 2). The haplotype network also showed that Hap 1 and Hap 2 were shared by 15 populations, Hap 13 was shared by 11 populations. In general, the haplotype network showed a shallow double star-shaped structure (Appendix A).

A total of 91 haplotypes were detected in 95 *D-Loop* sequences from the 15 populations, which were defined as Hap 1-91, consisting of one shared haplotype and 90 exclusive haplotypes. The frequencies of Hap 25, Hap 29, Hap 47 and Hap 72 were all 2.11%, and the rest of haplotypes were found only once. According to the geographic distribution of haplotypes, Hap 29 was shared by two populations (XM and DZG). There were few shared haplotypes between diverse populations and no dominant haplotypes (Figure 2). The haplotype network also showed that only Hap 29 was shared by two populations. Additionally, the haplotype network presented a bush-like shape (Appendix A).

### 3.3. Population Genetic Structure

Based on *COI* sequences, at the region level, −0.22% of the variation was found between regions, with −0.18% of the variance found within regions among populations and 100.40% within populations. At the ecoregion level, most of the total variation (100.04%) was accounted for by differentiation within populations, with a further −0.80% accounting for variation within ecoregions among populations, and the remainder (0.76%) partitioned among ecoregions (Table 3). At the population level, −0.29% of the total variance was found among populations, whilst 100.29% of the variance was found within populations. The AMOVA statistical tests were not significant in all three levels (*p* > 0.05). The pairwise population *F_ST_* showed that the genetic differences between populations ranged from −0.0456 to 0.0602. All pairwise *F_ST_* values were not statistically significant (*p* > 0.05; Appendix A).

Based on the *D-Loop* sequences, at the region level, −0.86% of the variation was found between regions, with −0.44% of the variance found within regions among populations and 101.29% within populations. At the ecoregion level, most of the total variation (101.27%) was accounted for by differentiation within populations, with a further 0.04% accounting for variation within ecoregions among populations, and the remainder (−1.31%) partitioned among ecoregions. At the population level, −0.82% of the total variance was found among populations, whilst 100.82% of the variance was found within populations (Table 3). AMOVA statistical tests in all three levels were not significant (Table 3). Furthermore, the *F_ST_* values between populations were distributed between -0.1128 and 0.2675, the genetic differentiation between the remaining populations showed insignificant differences (*p* > 0.05), except for the significant differences between MZD and FYLD, and between MZD and HA (*p* < 0.05; Appendix A). Two genetic markers revealed a high level of genetic homogeneity among the 15 populations of horn-eyed ghost crab and there was no significant genetic structure.

### 3.4. Migration and Connectivity

Based on *COI* sequences, the estimated historical gene flow ranged from 4.921 (MZD-FYLD) to 248.595 (QA-DH), while the estimated historical gene flow varied from 10.738 (SYH-DT) to 334.675 (CW-FYLD) based on *D-Loop* sequences. Two markers revealed the high historical gene flow among populations of horn-eyed ghost crab. The historical gene flow from MZD to most other populations was lower than that of most other populations to MZD based on two markers.

Based on *COI* sequences, the estimated migration rate ranged from 134.7 (QA-FYLD) to 390.1 (MZD-QA), while the estimated migration rate varied from 159.3 (PT-SYH) to 284.1 (SYH-NJLD) based on *D-Loop* sequences. The analysis consequences of two markers exhibited the high migration rates among populations. The *COI* sequences further showed the asymmetric migration rate with the higher migration rate from south to north than from north to south. Meanwhile, the migration rate from SYH to most other populations was higher than the migration rate from that of most other populations to SYH based on two markers (Figure 3). In addition, Migrate-n also revealed that the effective population sizes of each population based on both markers were relatively close (Appendix A).

## 4. Discussion

In this study, 15 populations of horn-eyed ghost crab were collected in or near 15 MPAs in the East and South China Seas to analyze connectivity among populations based on *COI* and *D-Loop*. The results revealed that the high genetic diversity and connectivity among populations, and a high historical gene flow and migration rate among 15 populations were supported by two markers. Additionally, the *COI* sequences further showed the asymmetric migration rate with a higher migration rate from south to north than from north to south. The results of population structure and connectivity could reflect the connectivities between the marine protected areas in the two seas. These results could offer information for the management of the marine protected areas in the study areas.

### 4.1. Differences of Genetic Diversity in Populations

High haplotype diversity (Hd > 0.5) and low nucleotide diversity (π < 0.005) were observed based on *COI* sequences when the 15 populations were considered as a whole. This low nucleotide diversity and high haplotype diversity were in line with the genetic diversity of *COI* sequences from other marine organisms, particularly marine crustacean species such as *Pachygrapsus crassipes* [28] and *Portunus trituberculatus* [29]. *D-Loop*, on the other hand, showed high haplotype diversity (Hd > 0.5) and high nucleotide diversity (π > 0.005). In contrast, the higher genetic diversity was revealed by *D-Loop* marker, which may be related to the mutation rate of the fragment [30]. *D-Loop* is the high mutation region, located on either side of the central conserved region, which evolves two to five times faster than mitochondrial protein-coding genes [31], and thus may be more sensitive in genetic diversity analysis than *COI* marker [32]. This disagreement was often seen in studies of multiple molecular markers [31].

Because the sampling sites of horn-eyed ghost crab were located in or near MPAs, the results of genetic diversity may provide some reference for the management effectiveness of MPAs. Species and populations are adversely affected by anthropogenic activities, such as habitat modification, which are a global phenomenon [33]. These negative impacts eventually would reduce the genetic diversity [34]. The management effectiveness of a protected area is generally considered to be closely related to its conservation effectiveness, and a good protection management is also expected to have a higher level of biodiversity [35]. In this study, we also found that the marine protected areas with high genetic diversity also had a higher management effectiveness. For example, the high genetic diversity in NJLD and MA was observed, for which a high management effectiveness in both protected areas was reported [36]. FYLD had the low genetic diversity based on two markers, and it also had a lower score of management effectiveness in this protected area [37]. Therefore, it might support that the level of population genetic diversity is related to the management effectiveness of the marine protected areas. However, more tests are required to explore the relationship between genetic diversity and MPA management effectiveness.

### 4.2. Connectivity Differences among Populations

The spatial distribution of haplotypes and the haplotype network based on *COI* sequences showed that the most common Hap 1 and Hap 2, accounting for almost 49.40% of samples, were found in all 15 populations. Shared haplotypes may be related with the large population sizes and high rates of migration [38]. Among the *COI* haplotypes, shared haplotypes existed between distant populations (up to 1000 km), which indicated that dispersal of this species may occur over long distances. In contrast, the existence of lower-frequency haplotypes was related with large population sizes, and this is because these haplotypes were not eliminated by the selection when in a large stable population [39]. Whereas the spatial distribution of haplotypes and the haplotype network from *D-Loop* revealed few shared haplotypes among populations, it may not represent low connectivity among populations, but be due to DNA superdiversity [40,41]. This result may be explained by the combination of small samples and high mutation rates in *D-loop* sequences [41]. Therefore, results from *COI* might provide more reliable results on the spatial distribution and connectivity.

In this research, the high level of connectivity among 15 populations, between ECS populations and SCS populations and among three ecoregion populations of horn-eyed ghost crab was observed based on the results of AMOVA and *F_ST_* for two markers, which was also consistent with the findings of *Collichthys lucidus* [42], *Larimichthys crocea* [43], *Pampus chinensis* [44] and *Thamnaconus hypargyreus* [17]. The high connectivity among populations may be due to the long planktonic larval period of 34–42 days for horn-eyed ghost crab [45], which could allow distant geographical distances between populations to be overcome and produce genetic homogeneity [46]. On the other hand, ocean currents may promote connectivity among populations [42]. During the planktonic larval stage of horn-eyed ghost crab in summer, the current from the surface to the bottom is almost consistent with the northeast flow parallel to the coast in the coastal area from Xiangshan to Pearl River Estuary [47], which may result in high connectivity among populations. Meanwhile, this may account for the higher migration rate from south to north revealed by the *COI* marker in this study. It was also in line with the results of an analysis of seven mangrove species on the southern coast of China [48]. Nevertheless, the directional difference was not observed in the migration rate results based on *D-Loop* sequences, which may be due to the fact that the higher mutation rate of the *D-Loop* marker was not conducive to the prediction of gene flow or less number of *D-Loop* sequences [49]. In addition, Migrate-n also revealed a large population size in each population of horn-eyed ghost crab. Such population may be insensitive to the loss and reorganization of variation by genetic drift, and could maintain the ancestral genetic information [50], leading to high genetic connectivity among populations.

### 4.3. Conservation and Management Implications for MPA

Although it is not clear whether marine protected areas in the East and South China Seas could form a network, the results of this study may provide some suggestions for the construction of such a network due to the sampling sites of horn-eyed ghost crab being located in or near protected areas. The higher migration rate from south to north related to the current in summer was revealed based on *COI* sequences in this research. Consequently, the hydrodynamic characteristics of the East and South China Seas should be considered in constructing a MPA network. In the near-shore areas of the East and South China Seas, the Guangdong coastal current (GDCC) and the coastal currents of Zhejiang and Fujian (MZCC) are the main currents [27]. In summer, the direction of MZCC and GDCC is northward along the parallel shoreline, but in winter, the MZCC and GDCC flow in the southwesterly direction [47]. These season-changed currents should be included in the design of the MPA network, because organisms that reproduce in different seasons may have completely different connectivity patterns [51], especially those that reproduce in only one season.

Many protected species such as *Acropora solitarvensis, Epinephelus akaara* and *Epinephelus bruneus* spawn during summer [52]. The summer fishing moratorium implemented by the Chinese government in the East and South China Seas [53,54] contributes to the protection of larvae and may complement the MPA network [55]. This has also been shown in previous studies. A study of fish resources in Daya Bay showed that the stock density, species number, biodiversity and evenness index increased after the summer fishing moratorium period, indicating that the structure of the ecological community improved [56]. Additionally, in a study in the coastal ocean of Ningbo, the SCS showed that species richness, abundance and biomass of macrobenthos communities increased significantly during the summer fishing moratorium period, which implied this policy could facilitate the resilience of microbenthic communities [57]. However, focusing solely on the spawning patterns of these species for reserve design will decrease protection of fall–winter spawners when the direction of the gyres and the location of upstream larval sources reverses [51]. For example, at least some important species such as *Anguilla japonica* and *Penaeus japonicus* also spawn exclusively during fall–winter [52]. So, flexible management strategies could be carried out during the winter for the conservation of these species. For example, seasonal closures or fishing intensity restrictions could be assigned to their spawning sites and core areas of activity [58]. Moreover, seasonal protection corridors could be set up in areas with high connectivity during the planktonic larval period, and these areas should be a priority for new protected areas in the future.

## 5. Conclusions

This study investigated the genetic connectivity among the populations of *Ocypode ceratophthalmus* in the East and South China Seas based on wide geographic sampling and two genetic markers. The results showed that the horn-eyed ghost crab had high genetic diversity. The AMOVA analysis revealed no significant genetic structure and high genetic connectivity among the populations. All the *F_ST_* values obtained from *COI* sequences were inapparent. The *F_ST_* values based on *D-Loop* sequences were only significant between MZD and FYLD as well as between MZD and HA. Migrate-n revealed the high historical gene flow and connectivity among 15 populations based on two markers. The *COI* sequences further showed asymmetric connectivity with higher connectivity from south to north than from north to south. The outcomes from this study provide suggestions for the construction of marine protected areas.

## Figures and Tables

**Figure 2 biology-12-00437-f002:**
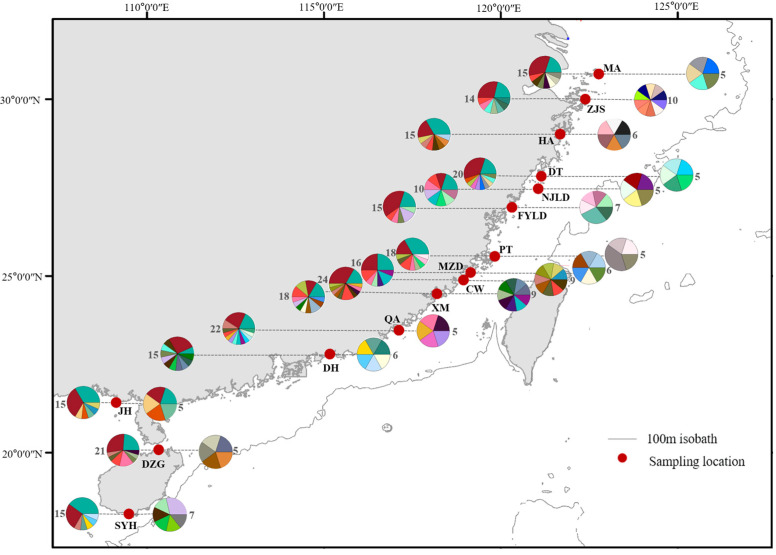
Haplotype map for horn-eyed ghost crab *COI* (**left**) and *D-loop* (**right**). Numbers represented the number of sequences used for analysis. Different colors represented different haplotypes.

**Figure 3 biology-12-00437-f003:**
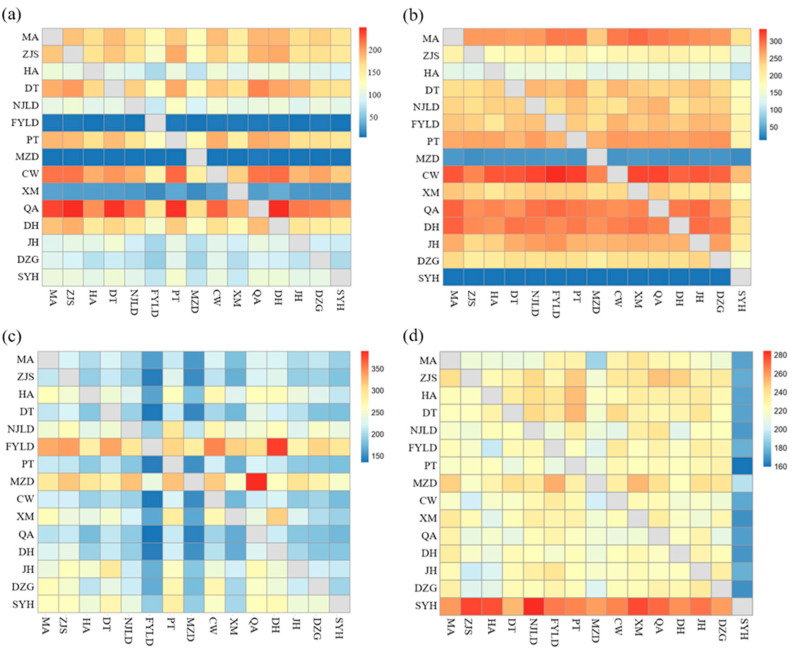
Historical gene flow (Nm) and migration rate (M) among 15 populations of horn-eyed ghost crab based on *COI* and *D-Loop* sequences. (**a**) Nm values based on *COI* sequences; (**b**) Nm values based on *D-Loop* sequences; (**c**) M values based on *COI* sequences; (**d**) M values based on *D-Loop* sequences.

**Table 1 biology-12-00437-t001:** Location and sampling details for each population.

Region	Ecoregion	MPA *	Population	Number of Samples	Weight (g)	Sampling Time
East China Sea	East China Sea Ecoregion	Dongtou National Marine Park (in)	DT	20	18.06 ± 6.17	27 April 2022
		Fuyao Archipelago National Marine Park (in)	FYLD	15	18.63 ± 5.70	30 March 2022
		Haitan Bay National Marine Park (in)	PT	18	13.68 ± 4.54	6 March 2022
		Hua’ao Island National Marine Park (near)	HA	15	15.89 ± 4.15	8 April 2022
		Ma’an Archipelago Special Marine Protected Area (in)	MA	15	5.41 ± 2.25	10 December 2021
		Meizhou Island National Marine Park (in)	MZD	16	13.38 ± 2.66	5 March 2022
		Nanji Archipelago National Nature Reserve (in)	NJLD	10	16.15 ± 6.95	18 April 2022
		Zhongjieshan Archipelago Special Marine Protected Area (near)	ZJS	14	22.82 ± 7.11	18 April 2022
	Southern China Ecoregion	Chongwu National Marine Park (in)	CW	24	16.59 ± 5.40	8 January 2022
		Xiamen National Marine Park (in)	XM	18	16.65 ± 5.72	2 January 2022
South China Sea	Southern China Ecoregion	Qing’ao Bay National Marine Park (near)	QA	22	15.01 ± 3.54	8 January 2022
		Zhelang Peninsula National Marine Park (near)	DH	15	14.43 ± 4.56	22 March 2022
	Gulf of Tonkin Ecoregion	Dongzhai Bay National Nature Reserve (near)	DZG	21	5.28 ± 2.51	18 September 2021
		Jinhai Bay Mangrove Protected Area (near)	JH	15	18.89 ± 6.09	1 April 2022
		Sanya River Mangrove Nature Reserve (near)	SYH	15	5.54 ± 2.50	18 September 2021

* In this column, the contents in parentheses indicate that the sampling site is in or near the marine protected areas, “in” indicates that the sampling site is in the reserve, and “near” indicates that the sampling site is near the marine protected areas.

**Table 2 biology-12-00437-t002:** Genetic diversity for 15 populations of horn-eyed ghost crab based on *COI*/*D-Loop* sequences. N, number of specimens for each population; h, the number of haplotypes; Hd, haplotype diversity; π, nucleotide diversity.

Region	Population	*COI*				*D-Loop*			
		N	h	Hd	π	N	h	Hd	π
The East China Sea	CW	24	12	0.87 ± 0.05	0.0034 ± 0.0006	9	9	1.00 ± 0.05	0.0279 ± 0.0033
	XM	18	12	0.95 ± 0.03	0.0054 ± 0.0007	9	9	1.00 ± 0.05	0.0281 ± 0.0028
	MA	15	9	0.88 ± 0.07	0.0037 ± 0.0007	5	5	1.00 ± 0.13	0.0314 ± 0.0052
	PT	18	10	0.88 ± 0.06	0.0050 ± 0.0010	5	4	0.90 ± 0.16	0.0271 ± 0.0050
	MZD	16	8	0.88 ± 0.05	0.0030 ± 0.0005	6	6	1.00 ± 0.10	0.0230 ± 0.0033
	HA	15	9	0.88 ± 0.07	0.0034 ± 0.0006	6	6	1.00 ± 0.10	0.0192 ± 0.0035
	NJLD	10	9	0.98 ± 0.05	0.0043 ± 0.0008	5	5	1.00 ± 0.13	0.0298 ± 0.0051
	ZJS	14	9	0.90 ± 0.06	0.0036 ± 0.0007	10	10	1.00 ± 0.05	0.0306 ± 0.0028
	FYLD	15	7	0.82 ± 0.08	0.0022 ± 0.0004	7	6	0.95 ± 0.10	0.0233 ± 0.0035
	DT	20	11	0.86 ± 0.07	0.0037 ± 0.0007	5	5	1.00 ± 0.13	0.0257 ± 0.0048
The South China Sea	QA	22	14	0.93 ± 0.04	0.0036 ± 0.0005	5	5	1.00 ± 0.13	0.0331 ± 0.0063
	JH	15	7	0.81 ± 0.08	0.0027 ± 0.0006	5	5	1.00 ± 0.13	0.0379 ± 0.0061
	DZG	21	9	0.86 ± 0.05	0.0027 ± 0.0004	5	5	1.00 ± 0.13	0.0268 ± 0.0064
	SYH	15	7	0.80 ± 0.08	0.0032 ± 0.0009	7	6	0.95 ± 0.10	0.0278 ± 0.0038
	DH	15	12	0.94 ± 0.05	0.0052 ± 0.0010	6	6	1.00 ± 0.10	0.0313 ± 0.0052

**Table 3 biology-12-00437-t003:** AMOVA results of horn-eyed ghost crab based on *COI* and *D-Loop* sequences.

Different Classifications	Genetic Markers	Source of Variation	df	Sum of Squares	Variance Components	Percentage of Variation
Region level (ECS and SCS)	*COI*	Between regions	1	0.708	−0.002	−0.22
		Within regions among populations	13	12.383	−0.002	−0.18
		within populations	238	233.873	0.983	100.40
		Total	252	246.964	0.979	
	*D-Loop*	Between regions	1	5.481	−0.072	−0.86
		Within regions among populations	13	108.307	−0.037	−0.44
		within populations	80	685.181	8.565	101.29
		Total	94	798.968	8.455	
Ecoregion level (East China Sea Ecoregion, Southern China Ecoregion, Gulf of Tonkin Ecoregion)	*COI*	Among ecoregions	2	2.857	0.007	0.76
		Within ecoregions among populations	12	10.234	−0.008	−0.80
		within populations	238	233.873	0.983	100.04
		Total	252	246.964	0.982	
	*D-Loop*	Among ecoregions	2	10.766	−0.111	−1.31
		Within ecoregions among populations	12	103.021	0.003	0.04
		within populations	80	685.181	8.565	101.27
		Total	94	798.968	8.457	
Population level	*COI*	Among populations	14	13.091	−0.003	−0.29
		within populations	238	233.873	0.983	100.29
		Total	252	246.964	0.980	
	*D-Loop*	Among populations	14	113.787	−0.070	−0.82
		within populations	80	685.181	8.565	100.82
		Total	94	798.968	8.495	

## Data Availability

DNA sequences have been deposited in GenBank.

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
