# Peer review of "Genetic Diversity and Connectivity of Ocypode ceratophthalmus in the East and South China Seas and Its Implications for Conservation"

_biology, 2023, doi:10.3390/biology12030437_

Round 1

Reviewer 1 Report

Comments to the Author

Connectivity of marine protected areas in the East and South China Seas: a case study of horn-eyed ghost crab (Ocypode ceratophthalmus)

Authors

Feng Zhao , Yue Liu , Zihan Wang , Jiaying Lu , Ling Cao , Cong Zeng

I think that the study deals with a very interesting and actual topic, related to the network of MPAs for preserving marine biodiversity. But it is not clear to me which is the main aim of this paper: I don’t understand how the study of a species with high dispersal ability can support the establishment of a MPA network; I think that it could be rather the contrary (a network can favour the connectivity among populations and a posteriori you should examine the effects of that management). A species with planktonic larvae has usually low genetic differentiation independently from MPA network (authors at rows 309-311 say that the consequences of connectivity must be interpreted with caution ought to high self recruitment…). On the other hand other species show genetic differentiation: “inconsistent findings” have been found and authors observed that whether .. row86-87. MPA network can be constructed providing ecological corridors or other strategies supporting connectivity between populations. As a consequence, I suggest to modify the aim and the title. In my opinion, the paper can be a good population genetic study that is performed in an area characterized by protected areas. Just it!

In addition, I would like to suggest:

-a reference is missing at row 60

-the paragraph at rows 100-111 is useless: I would put a table with pops labels N etc, that is a part of the table S1

-row 121: the appropriate amount is meaningless, provide the exact amount

-no explanations for the different size of obtained sequences for the two gene portions was provided: 253 vs 95…

-a network (a minimum spanning tree or a median joining) must be provided

-what is the meaning of the paragraph from 283 to 289?

Reviewer 2 Report

Dear Authors,

I have carefully reviewed the submission about the genetic structure of horn-eyed ghost crabs in f marine protected areas in the East and South China Seas. They found that horn-eyed ghost crabs had high genetic diversity and high connectivity among populations in the study area. The species had a high historical gene flow and migration rate among the studied populations for COI and D-Loop markers. 

The results were well-present, with tables and figures in detail. The text was written as brief and concise.  I only suggest minor comments in the text. In my opinion, the submission should be published after minor revisions.

Best regards,

Reviewer 3 Report

Review

Paper title: Connectivity of marine protected areas in the East and South China Seas: a case study of horn-eyed ghost crab (Ocypode ceratophthalmus)

The authors collected samples at 15 locations in the East and South China Seas to reveal if there is connectivity among populations of the horn-eyed ghost crab Ocypode ceratophthalmus. They found high genetic diversity and connectivity among the populations as a result of high historical gene flow and migration rates. The authors concluded that the migration rate in the northern direction was higher than in the southern direction. This study expands our knowledge of the connectivity of marine protected areas in the region.

All these reasons explain the relevance of the paper by Feng Zhao and co-authors submitted to "Biology".

General scores.

The data presented by the authors are original and significant. The study is correctly designed and the authors used appropriate sampling methods. In general, statistical analyses are performed with good technical standards. The authors conducted careful work that may attract the attention of a wide range of specialists focused on biogeography.

Recommendations.

The authors should provide more information about the crab specimens (size, weight, sex), tissue collection procedures.

L 113. “morphological identification”. Please, provide a source for a guide of identification used.

Provide information about congener species in the region.

Specific remarks.

L 9. Consider replacing “the high connectivity” with “high connectivity”

L 25. Consider replacing “diversities, the consequences of” with “diversities,”

L 28. Consider replacing “migration rate” with “migration rates”

L 30. Consider replacing “higher migration rate” with “a higher migration rate”

L 41. Consider replacing “was regard” with “is regarded”

L 50. Consider replacing “in 20  years ago by Chinese government” with “20  years ago by the Chinese government”

L 60. Consider replacing “Convention” with “The Convention”

L 70. Consider replacing “effected” with “affected”

L 71. Consider replacing “environment” with “environments”

L 75. Consider replacing “were reported” with “reported”

L 76. Consider replacing “studies reported” with “studies that reported”

L 77. Consider replacing “two seas” with “the two seas”

L 97. Consider replacing “supports” with “support”

L 151. Consider replacing “to estimated” with “to estimate”

L 193. Consider replacing “from 15 populations” with “from the 15 populations”

L 205. Consider replacing “At population level” with “At the population level”

L 207. Consider replacing “At region level” with “At the region level”

L 209. Consider replacing “At  ecoregion  level” with “At  the ecoregion  level”

L 217. Consider replacing “At population level” with “At the population level”

L 218. Consider replacing “At region level” with “At the region level”

L 220. Consider replacing “At  ecoregion  level” with “At  the ecoregion  level”

L 228. Consider replacing “the high level” with “a high level”

L 242. Consider replacing “the high migration rates” with “high migration rates”

L 255. Consider replacing “to analyzed” with “to analyze”

L 261. Consider replacing “the high historical gene flow and migration rate among 15 populations was” with “a high historical gene flow and a migration rate among 15 populations were”

L 263. Consider replacing “higher migration rate” with “a higher migration rate”

L 270. Consider replacing “genetic diversity of COI marker” with “the genetic diversity of COI markers”

L 274. Consider replacing “fragment” with “the fragment”

L 278. Consider replacing “in the studies” with “in studies”

L 281. Consider replacing “often associated” with “is often associated”

L 287. Consider replacing “related with” with “related to”

L 288. Consider replacing “However, it was still needed more evidences” with “However, more efforts are needed”

L 304. Consider replacing “haplotypes sharing” with “haplotype sharing”

L 319. Consider replacing “by COI marker” with “by the COI marker”

L 321. Consider replacing “in the southern coast of China [47]. Nevertheless, directional” with “on the southern coast of China [47]. Nevertheless, the directional”

L 323. Consider replacing “rate of D-Loop” with “rate of the D-Loop”

L 342. Consider replacing “is feasible is” with “is feasible and”

L 352. Consider replacing “current of” with “current in”

L 357. Consider replacing “are northward” with “is northward”

L 358. Consider replacing “in  southwesterly direction” with “in the southwesterly direction”

L 359. Consider replacing “into the design” with “in the design of”

L 365. Consider replacing “of larva” with “of larvae”

L 368. Consider replacing “structure of ecological community improved” with “the structure of the ecological community improved”

L 380. Consider replacing “the areas with high connectivity during” with “areas with high connectivity during the”

L 382. Consider replacing “the comprehensive view” with “a comprehensive view”

L 383. Consider replacing “on widely” with “on wide”

Round 2

Reviewer 1 Report

I don't agree with the title and I keep on my previous considerations about the aim of the paper. Authors added minimum spanning network but no commente were presented along the paper, no comments related to figures S2...

Round 3

Reviewer 1 Report

Authors modified the title and the text accordingly with my suggestions